# DISCO: A deep learning ensemble for uncertainty-aware segmentation of acoustic signals

**Thomas Colligan[1,2☉], Kayla Irish[2,3☉], Douglas J. Emlen[4], Travis J. Wheeler[1,2]***

**1** College of Pharmacy, University of Arizona, Tucson, AZ, United States of America, **2** Department of Computer Science, University of Montana, Missoula, MT, United States of America, **3** Department of Statistics, University of Washington, Seattle, WA, United States of America, **4** Division of Biological Sciences, University of Montana, Missoula, MT, United States of America

☉ These authors contributed equally to this work.
\* twheeler@arizona.edu

## Abstract

Recordings of animal sounds enable a wide range of observational inquiries into animal communication, behavior, and diversity. Automated labeling of sound events in such recordings can improve both throughput and reproducibility of analysis. Here, we describe our software package for labeling elements in recordings of animal sounds, and demonstrate its utility on recordings of beetle courtships and whale songs. The software, DISCO, computes sensible confidence estimates and produces labels with high precision and accuracy. In addition to the core labeling software, it provides a simple tool for labeling training data, and a visual system for analysis of resulting labels. DISCO is open-source and easy to install, it works with standard file formats, and it presents a low barrier of entry to use.

**Data Availability Statement:** DISCO software can be installed using pip (https://pypi.org/project/disco_sound/). Source code is available at https://github.com/TravisWheelerLab/disco and material

## 1 Introduction

Animals produce an astonishing diversity of sounds, ranging from alarm calls and courtship songs to echolocating sonar [1–3]. For example, male field crickets (Teleogryllus oceanicus) produce acoustic signals using specialized structures on their wings, which can attract both conspecific females and deadly parasitic flies [4]. Biologists have documented dramatic and rapid changes to this signal, including the complete loss of the song [5] and the origin of an entirely new song that "purrs" rather than "chirps" [6]. Studies such as this require biologists to analyze field recordings of animal sounds, and to isolate the elements associated with the signal—chirps or purrs, for example –from background sounds also included in the recording. Decades of animal communication research have refined methods for recording and analyzing animal sounds [7–10], but current methods have significant limitations [11–13]. Animal sound analysis typically relies on a necessary pre-processing step that involves the segmentation of recordings to delineate classes of sound events. Traditionally, this process has relied on manual annotation of sound events guided by a visual representation (i.e. spectrogram) of the recordings. Along with being time-consuming, manual sound annotation introduces errors due to variability in the recording conditions (e.g., background noise [3, 14]) subjectively

(data + code) to reproduce manuscript figures is available at https://osf.io/r2nfu/.

**Funding:** We also acknowledge funding from the National Institute of General Medical Sciences (NIGMS), National Institutes of Health (NIH) GM132600, and the Division of Integrative Organismal Systems (IOS), National Science Foundation (NSF) 2015907. The funders had no role in study design, data collection and analysis, decision to publish, or preparation of the manuscript.

**Competing interests:** The authors have declared that no competing interests exist.

selected brightness and contrast settings when observing the spectrogram [3] and, especially, inconsistency in observer judgment [3, 8, 11, 12]. More recently, computational methods, particularly in machine learning contexts, have set the path towards improved automation and classification of diverse acoustic signals. Examples of recent machine learning tools intended to automate acoustic signal labeling include SongExplorer [15] and DeepSqueak [16]. DeepSqueak relies on Fast Region-based Convolutional Networks [17] to classify mouse ultrasonic vocalizations, treating spectrograms as images. SongExplorer uses a small Convolutional Neural Network (CNN) with pooling to classify individual fruit fly sound pulses, tiling subsequent predictions to get dense sequences of labels.

Although existing computational methods remove the burden of manual annotation and reduce inconsistencies in the classification of sound events, model training remains challenging, and existing methods fail to produce uncertainty estimations for those classifications. Here, we present DISCO (DISCO Implements Sound Classification Obediently), software that simplifies model training, automates the process of classifying sound events, and provide accurate estimations of classification uncertainty. DISCO combines the properties of state-of-the-art semantic segmentation methods (U-Net, [18]), model ensembling [19, 20], uncertainty quantification, and hidden Markov model post-hoc analysis to improve the precision and accuracy of automated signal labeling. Specifically, the incorporation of a model ensemble and a smoothing hidden Markov model (HMM) improves robustness to outliers and noise.

DISCO includes a tool for efficient manual labeling of training data, and it maintains format compatibility with the RAVEN .csv format [21] to provide functionality throughout the sound event annotation process. DISCO can be installed via `pip`, and is released under an open source license to promote its integration with biological analysis pipelines and encourage feature enhancement from the community.

To demonstrate the utility of DISCO, we primarily apply our method to a novel dataset of recordings of Japanese rhinoceros beetle (Trypoxylus dichotomus) chirps. We also demonstrate the general nature of the tool by labeling snippets of Right whale song recordings. Due to its user-friendly model training interface and accurate sound event classification with meaningful uncertainty estimates, we anticipate that DISCO will enable improved and simplified automated classification of sound events within recordings of animal sounds.

## 2 Sound data and labeling

### 2.1 A general introduction to sound data

Sound data consists of records of the amplitude and frequency of noise. These noises can be manufactured or captured in natural environments with a microphone. Classification of sound fragments typically first involves computing a spectrogram from the sound data, achieved through application of a Fast Fourier Transform (FFT) [22]. The result is a 2-dimensional matrix, with each column corresponding to a time point, each row corresponding to a frequency, and the value at a given cell in the matrix corresponding to the amplitude of a certain frequency at a certain time. This can be visualized as an image, examples of which are shown in Fig 1. Computing a spectrogram requires parameterization of FFT window size and overlap (for details, see [23]); defaults in DISCO are expected to perform well across near-audible frequencies, but may be adjusted through a configuration file. Fig 1 shows spectrograms formed from three different biological sounds: one of the male authors saying "hello", a bird song (Western Meadowlark, sound data downloaded from [24]), and a Japanese rhinoceros beetle chirping. The spectrograms were computed with the same window size and overlap and a log base 2 was applied after spectrogram computation for visualization. Each subplot in Fig 1

## hello　western meadowlark　beetle chirp

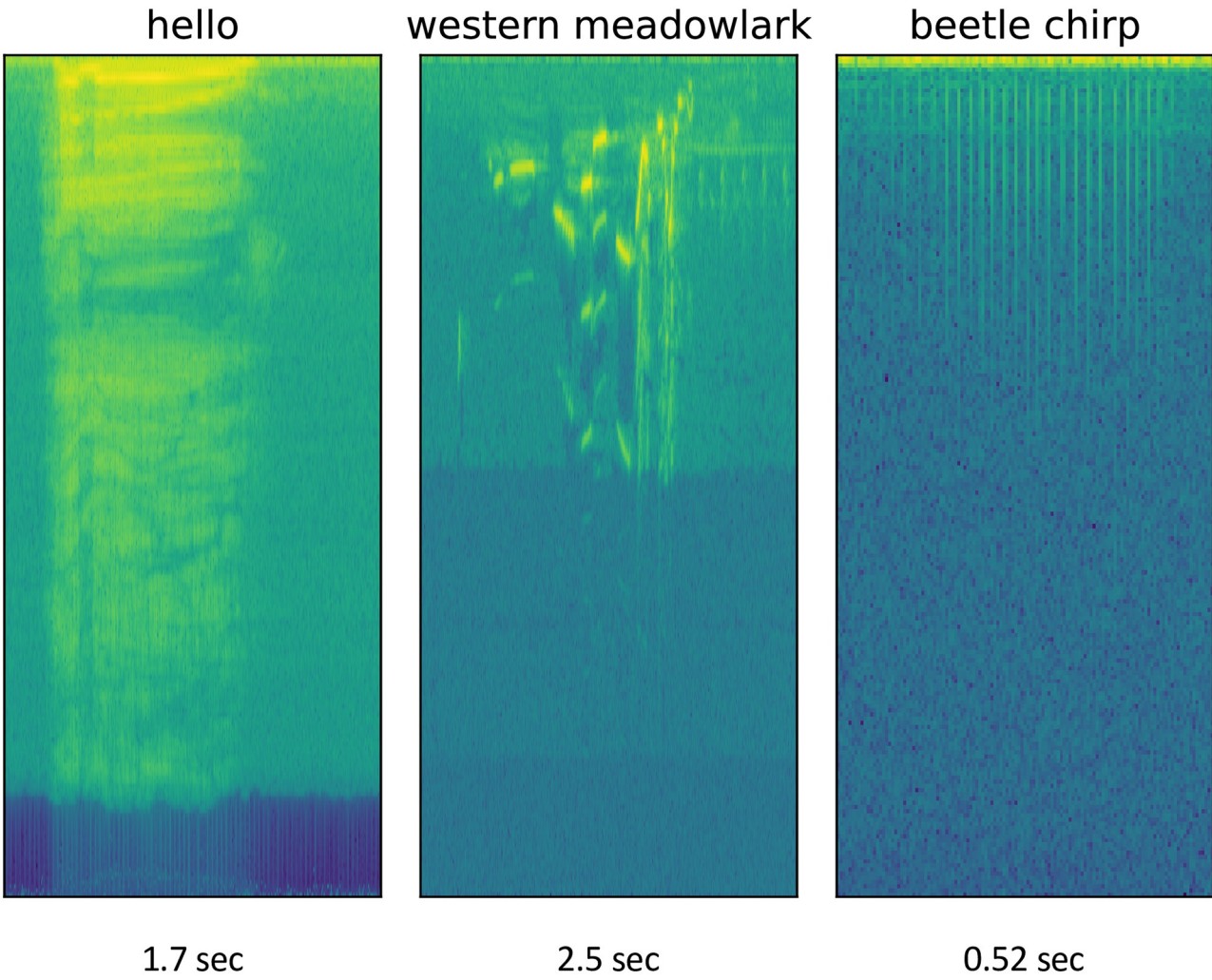

1.7 sec　　　　2.5 sec　　　　0.52 sec

**Fig 1. Spectrograms formed from three recordings: A voice saying "hello", a bird singing, and a beetle chirping (our dataset).** Equal image size does not indicate equal recording length: "hello" is 1.7s, the Western Meadowlark song 2.5s, and the beetle chirp 0.52s. In each spectrogram, values are reported with time increasing on the horizontal axis, and frequencies increasing up the vertical axis. Intensity of sound is presented at each time/frequency pairing, with low intensity presented as black, and high intensity as bright green.

is a different length: "hello" is 1.7s, the Western Meadowlark song 2.5s, and the beetle chirp 0.52s. We consider a dataset of beetle chirps in this analysis. Each of the subplots in Fig 1 shows unique events localized in time through the spectrogram, surrounded by "background" noise, which is noise that is not the main subject of the recording. Each type of sound is unique: the "hello" is spread out across a large frequency range. The bird call jumps between frequencies quickly, and the beetle chirp (due to the physical mechanism of generation) contains relatively uniform frequency content in a unique temporal pattern. Annotating where, and for how long, different sound events (noises) occur is a primary goal of many sound analysis tools, and effective analysis requires considering the temporal patterns and frequency content of the sound. For example, a user might want to label the location of all of the occurrences of "hello" in their recording, or similarly all beetle chirps of a given type. Analyzing temporal patterns of noises in recordings can provide insight into behavior, such as response to stimulus, or animal robustness.

## 2.2 Beetle sound data—Acquisition and processing

The initial motivation for developing DISCO was to annotate chirps of Japanese rhinoceros beetle (*Trypoxylus dichotomus*) courtship songs. Beetles were recorded in 4' x 4' x 4' boxes lined with anechoic studiofoam wedges to insulate the animals from ambient sounds like ceiling fans. They were recorded with a Sennheiser ME62 omnidirectional microphone and K6 power module. Recordings were single-channel and had a sample rate of 48Khz. Females were placed on upright logs inside the boxes with a 1" diameter x 1" deep hole drilled in the side and filled with jello. After females were settled and feeding, males were introduced to the base of the tree trunk in the arena and they crawled up to the female and (usually) climbed onto her back and began a series of trembling dances and stridulatory songs. The room was kept hot (about 30 C) since beetles were most active at these temperatures. Strings of red-filtered LED lights provided illumination for the researchers and emulated night-time, since Japanese rhinoceros beetles are nocturnal and cannot see red light. Despite best efforts at noise canceling, there was generally ambient sound picked up in each recording, e.g. a humming air conditioner (A/C), or the sounds of researchers shifting positions or talk-ing in the background.

The dataset contains two kinds of *T. dichotomus* chirps: the A chirp and B chirp. The A chirp is generated by retracting and extending the abdomen in a front-to-back motion. The B chirp is a result of side-to-side and back-and-forth sweeps of the abdomen against each elytral tip in alternation. Several chirps of a single kind will appear in a row during courtship, called "runs" throughout the rest of this paper. DISCO's focus was to annotate and classify each occurrence of a chirp (referred to as a "chirp", "event", or "sound event") in the recordings. The dataset also contains "background" or "X" sounds. This refers to anything not classified as an A or a B chirp—A/C hum, doors opening and closing, human voices, and the lack of a spe-cific sound event.

The first step of DISCO analysis is computation of a Mel-transformed spectrogram. The perceptual difference between 500 and 1000 Hz is much larger than the difference between 9,000 and 9,500 Hz [25]. The Mel scale [26] is a map that seeks to solve the perceptual differ-ences in human hearing: Mel units are equally spaced apart in pitch, which means differences in the Mel scale are magnitude-independent. The Mel-transformed spectrogram is used as the DISCO default spectrogram as it was observed to increase accuracy. DISCO also applies a high-pass filter to the sound data to remove low-frequency background noise. By default, this is performed by removing the first 20 frequency components from the Mel spectrogram, effec-tively truncating the height of each spectrogram; this setting can be controlled by DISCO con-figuration file.

DISCO relies on the transformed spectrogram for a variety of tasks, including users gener-ating labels. A software tool for labeling sound elements is contained within DISCO (see next section), and was used to annotate A, B, and background sound events. An example in spectro-gram space of A and B chirps is shown in Fig 2. Each subplot displays 6.25s of recording and contains runs of chirps of each type. The same audio recording contained both runs of chirps. B chirps tend to be longer than A chirps, with a mean length of 0.42 seconds compared to 0.14 seconds.

## 2.3 A simple tool for labeling sound elements

DISCO contains a lightweight and customizable tool for labeling sound elements. Alternative sound labeling tools often require multiple clicks to save an annotated example—for example, selecting a region and then selecting a tick box, and clicking a "save" button. DISCO instead relies on keypresses and mouse actions. To annotate a region of a recording, a user clicks-and-

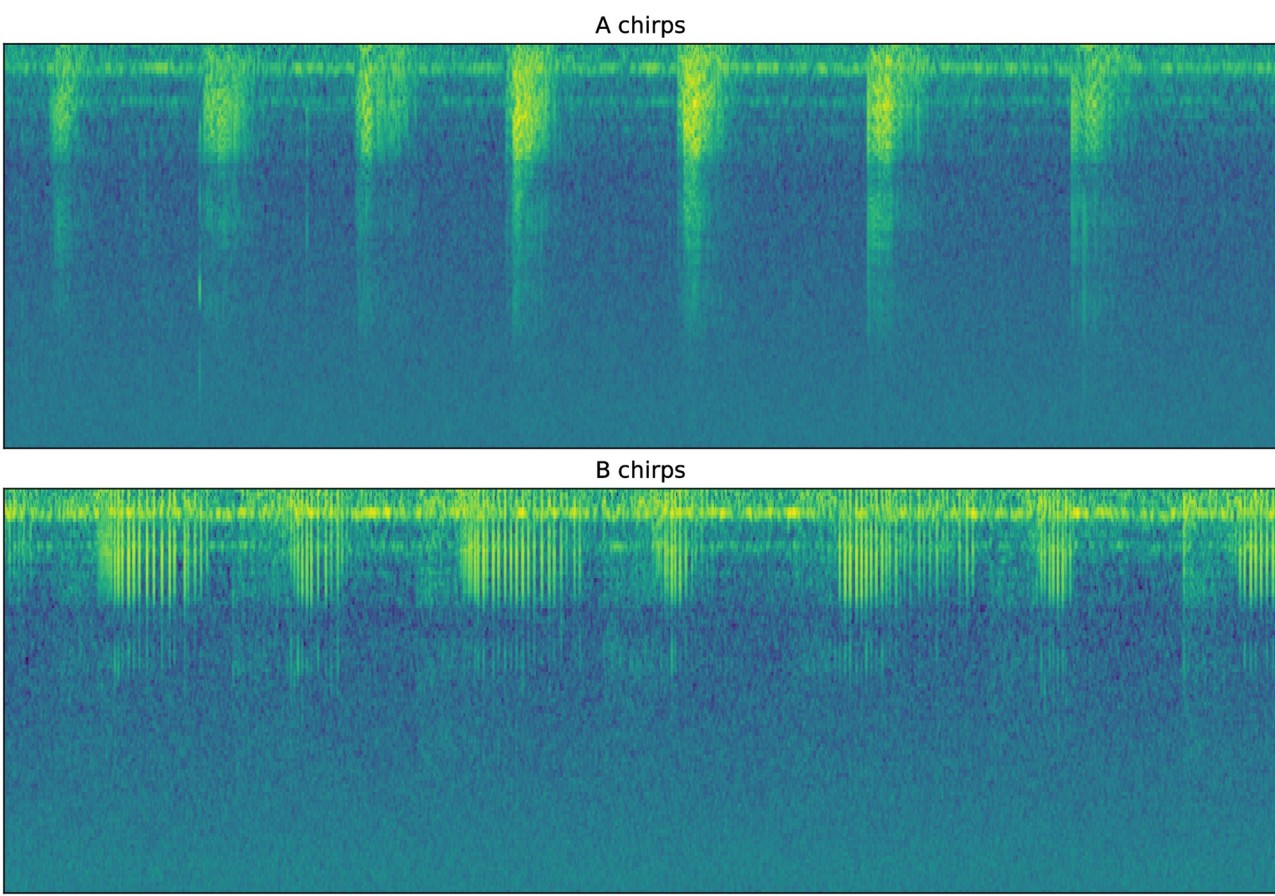

**Fig 2.** A and B sample chirps. Each subplot is 6.25s long and shows a run of the same type of chirps.

drags a box on the desired region. A zoom-in of the box is shown in a separate panel for quality assurance. The user then presses a key specified in a configuration file to save the annotated region as a specific class. Users can easily specify their own key mappings through a `python` dictionary. The labeling app can accommodate an arbitrary number of sound classes. A screen capture of the app is shown in Fig 3.

# 3 Design and implementation

## 3.1 Core model architecture

At its heart, DISCO is a tool for temporal segmentation of noises in spectrograms, and employs a U-Net based architecture [18] as its base model architecture. U-Nets are a class of convolutional neural networks designed for high-resolution multi-class image segmentation. The classical 2-dimensional (2-D) U-Net combines a downsampling and upsampling path to perform image segmentation. The combination of paths allows for high- and low-level concepts to be passed through the network with minimal information loss. A spectrogram is effectively a 2-D image, so a 2-D U-net seems to be a natural choice for segmentation. However, using a 2-D U-net with conventional 3x3 kernels would force the 2-D model to learn that signals along the frequency axis are related for a given timepoint. In contrast, a 1-D U-net allows direct enforcement of the fact that a given timepoint is associated with a long vector of frequency

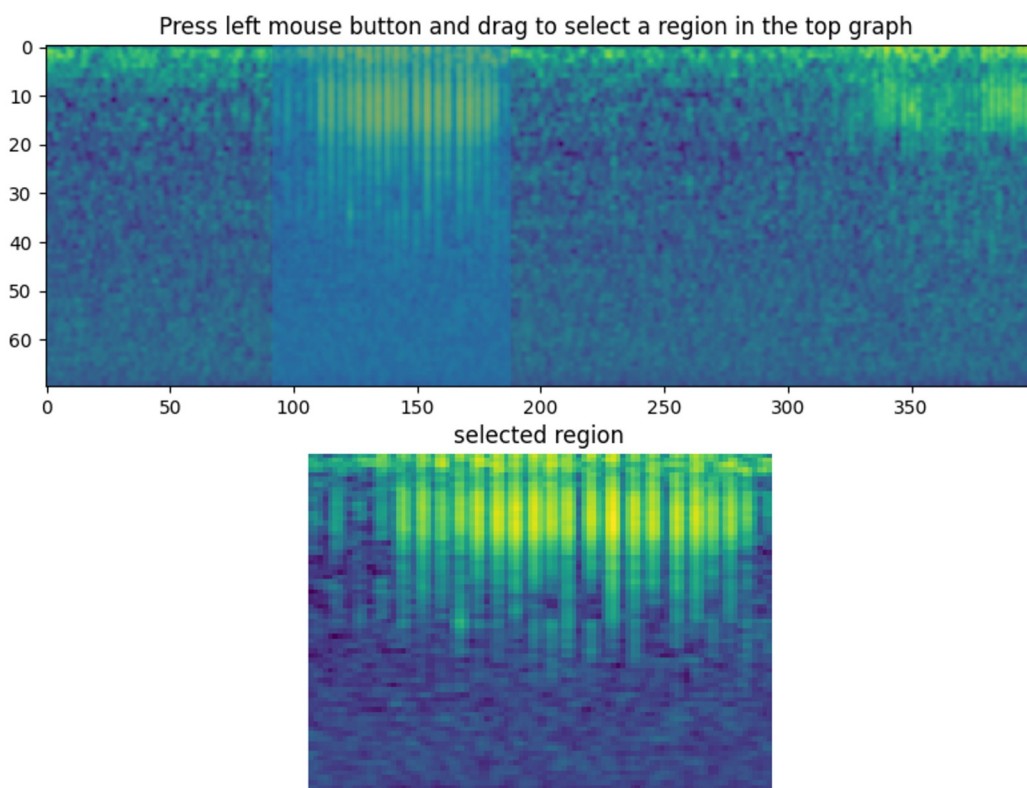

**Fig 3. A screenshot of the labeling app.** In the top spectrogram, the user is in the middle of a click-and-drag action around time points 90-180, resulting in the blue bounding box in that region. The bottom spectrogram shows a zoomed-in representation of that selected region.

information. As such, a 1-D U-net is a better match for temporal segmentation, and serves as the basis of segmentation in DISCO. The design of the 2-D U-net easily translates to segmentation of 1-D temporal signals, and DISCO implements a 1-D variant by replacing all 2-D convolutions with 1-D convolutions. DISCO by default uses a version of the U-Net with 1M parameters by reducing the number of convolutional filters in each layer relative to the implementation in [18].

Many applications of convolutional neural networks to sound data treat the spectrogram explicitly as a 2D image. In contrast, each DISCO 1-D U-Net treats a spectrogram as a sequence of time-oriented observations, and applies convolutions along the time axis. A large convolutional kernel is used along the frequency axis of the spectrogram in order to simultaneously combine all frequency information. Consider, for example, the differences in A and B beetle chirps; distinguishing between chirps does not require vertical spatial information—the most telling difference between A and B chirps is the temporal pattern of multi-frequency

pulses. Though this pattern holds for some sound data, other sound data (like bird song) can occupy unique frequency bands, so that localization in both time and frequency is important for classification. To support labeling of such data, DISCO also implements a 2-D U-Net, which can be enacted through configuration settings.

The beetle dataset consists of three sound types: A, B, and background. The U-Net architecture has a final three-way softmax layer, designed to assign each time point to one of those three classes. The softmax layer produces three floating point numbers that sum to one for each time point in the spectrogram. Softmax values for a given point can be interpreted as the probability the model assigns to the time point as belonging to each specific class. The number of classes (and accordingly the size of the softmax layer) can be controlled via a configuration file.

## 3.2 An ensemble of classification models

Sound recordings are often noisy and can contain anomalous sounds, leading to a level of error and uncertainty. Though softmax values serve as a form of labeling confidence, better calibrated estimates of predictive uncertainty are produced by aggregating the results of an ensemble of models [19, 20]. Moreover, the collective decision produced by an ensemble is less likely to be in error than the decision made by any of the individual networks [27, 28].

Specifically, a model ensemble is a collection of models that each produce an independent prediction for a given set of data. Each member of the ensemble learns a unique representation of signals in the dataset during training. Low-quality or noisy signals will be handled differently by each member of the ensemble, as each has been trained with differently initialized weights and/or a slightly different subset of the dataset. Uncertainty can be estimated by comparing the predictions of each member—high-fidelity signals will have concordant predictions while low-fidelity or uncertain predictions will differ between members of the ensemble.

For this reason, DISCO evaluates a sound recording with an ensemble of models, rather than applying a single trained U-net model. Two alternative ensembling techniques are implemented in DISCO: bootstrap aggregation (bagging) and a technique that we call "random initialization". Bagging [20] trains multiple models on different parts of the same dataset. Each model is trained with a subset of the dataset computed by sampling from the original dataset randomly with replacement until the size of the original dataset is reached. We call models trained with this ensembling technique "bootstrap" models, and each bootstrapped model uses a different initial set of randomly initialized weights. "Random initialization" (*random init* hereafter) relies simply on the random initialization and non-linear training of neural network weights to encode different biases [19]—each random init model is initialized with a different random seed and trained on the full training dataset. In bagging, each model is exposed to different facets of the entire dataset and may learn different labeling rules due to the unique biases encoded in the subsampled set. In random init, differences in ensemble members are entirely due to the stochastic nature of initialization and training. To gain insight into the utility of these alternative ensemble training strategies, we test six different ensembles: random init and bootstrap, testing each with member count of 2, 10, and 30.

**3.2.1 Classification and uncertainty quantification with the ensemble.** For each time point, DISCO aggregates softmax values from ensemble models in two ways: it computes both a median softmax value and a measure of softmax variability. To arrive at a class prediction for a given point, DISCO computes the median softmax value over the members in the ensemble, and the class with the highest median softmax is used as the prediction of the

ensemble. Uncertainty is estimated from the collection of softmax predictions via the computation of inter-quartile range (IQR), which computes the difference in softmax value between the $75^{th}$ percentile softmax value among ensemble members and the $25^{th}$ percentile softmax value. IQR effectively encodes different situations of ensemble predictions—an ensemble of 10 members in which 8 share very similar softmax values will have high certainty even if the remaining two members have discordant predictions. Each time point has three IQRs, one for each class. The IQR of a time point is the one for the predicted class. If IQR is above the given threshold, then the time point defaults to a background classification. The incorporation of IQR allows DISCO users to base predictions on an uncertainty threshold. Desired precision/recall trade-offs can be reached via IQR tuning. IQR ranges from 0 to 1 as do the softmax values.

**3.2.2 Evaluation of spectrograms.** DISCO can ingest and predict arbitrarily long sequence. Instead of predicting long sound recordings in one forward pass DISCO splits up the input of novel sound files to ease computational burden. Seamless predictions for the test files are computed using a 1-D version of the overlap-tile strategy proposed in [18].

## 3.3 Analysis setup

For beetle chirp labeling experiments, train and validation labels were collected differently from test labels. For train/validation data, spectrograms of beetle recordings were labeled by multiple different annotators using a combination of DISCO's labeling tool and the RAVEN software suite. The train and validation labels contained a considerable amount of label noise due to differences in annotator precision. In particular, the beginnings and ends of chirps are often difficult to precisely identify since they may look like background noise or degraded versions of the true chirp. Different annotators may have included or excluded low-confidence parts of chirps, effectively extending or truncating the chirp in time.

Because recordings may have had different background sound conditions or beetle chirp volume, annotation of training/validation data was collected across multiple recordings, so that the training set included a diverse range of sounds. The resulting annotations were split into train and validation sets by randomly partitioning the labeled sounds into two sets containing 88% and 12% of total. Model weights were trained with the Adam [29] optimizer for 100 epochs. Some hyperparameters—the number of FFTs in the spectrogram calculation, high-pass filter frequency, and learning rate—were optimized using an in-house implementation of the Hyperband [30] algorithm. These hyperparameters were only selected once, and then set as defaults during model training. The model's performance on the validation set was computed at the end of each epoch, and a snapshot of model weights was saved. At the end of the 100 epochs, the set of model weights that had the highest validation performance was kept as a set of weights used in the ensemble. This process was repeated once for each desired ensemble member.

The test set was labeled by a single annotator with instructions to (1) establish consistent rules about beginnings and ends of chirps and (2) to only annotate regions of chirps for which the annotator was certain of the label (this high-confidence labeling was also a guiding principle for training data). To ensure complete independence between train and test set, the test set was gathered from three recordings that were not in the train or validation set. It contains 1757 examples—667 A chirps and 1090 B chirps. Though training data consisted of short sound clips, each test file was a continuous recording of sound, and every time point was labeled as belonging to one of the three target classes. This replicates the intended use of DISCO as a tool for annotating long recordings.

### 3.4 Label smoothing with a hidden Markov model

The ensemble of CNNs in DISCO is not constrained to output continuous predictions. Despite being very accurate, the ensemble occasionally produces erroneous predictions—for example, a long region of "background" prediction punctuated by unrealistic single time point predictions of A or B chirps. The inverse can also occur, in which A or B chirps are split by short regions of background predictions. DISCO can optionally apply a hidden Markov model (HMM) to the predictions of the ensemble, to smooth unlikely interruptions in DISCO's predictions. The HMM architecture is simple, and controlled by configuration; a template and example are provided in the documentation.

The HMM used for beetle chirps encodes the transition rules that were observed during labeling. For example, A and B chirps can never be directly adjacent to one another due to the physical mechanism of generation. The HMM is shown in Fig 4. The application of the HMM was observed to aid qualitative performance.

### 3.5 Software engineering

DISCO is implemented in `python3`, specific dependencies on the Torchaudio library (for audio processing), the PyTorch library (for neural networks), and the pomegranate library (for

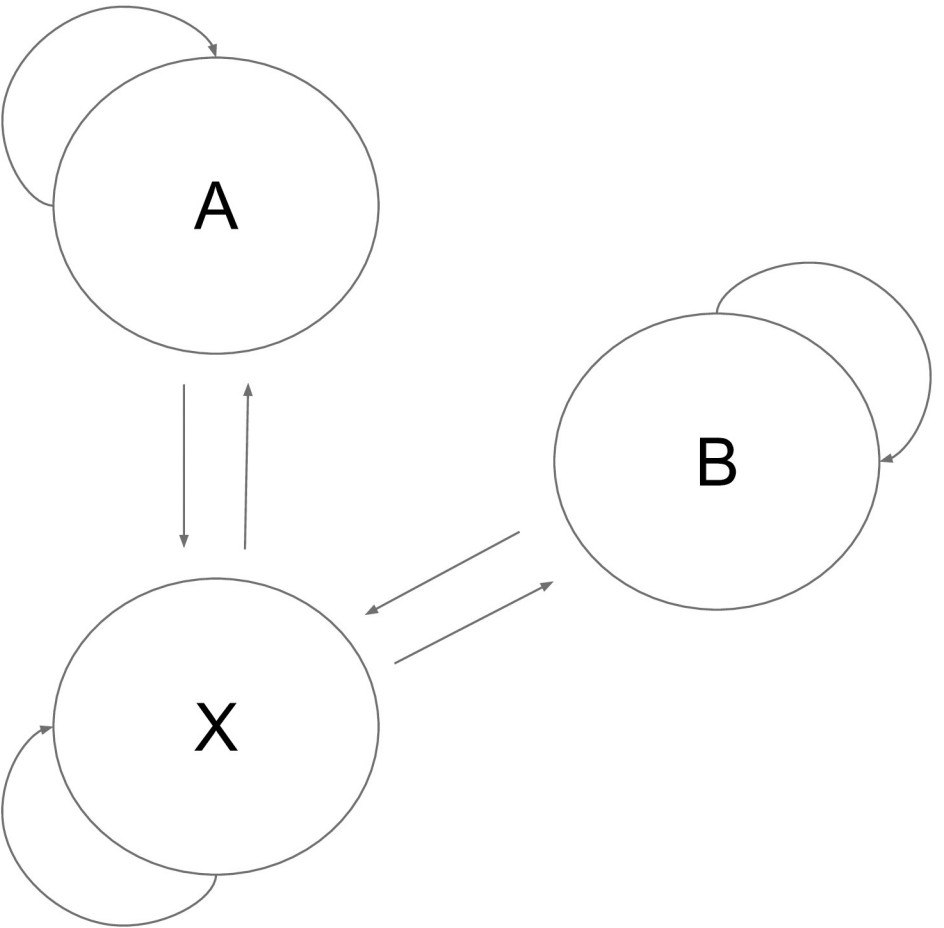

**Fig 4. Diagram of the HMM used to smooth data.** The X state corresponds to all background sounds.

hidden Markov models). It is accessible via a pip install or github, and comes with six command-line utilities: `disco (label, extract, shuffle, train, infer, visualize)`. Input and output labels of sound recordings are in the format accepted by Cornell's RAVEN software. Thorough documentation on how to use DISCO is available on github (https://github.com/traviswheelerlab/disco), and DISCO is distributed with a BSD 3-clause license.

## 4 Results

### 4.1 Measuring accuracy

**4.1.1 Data input, accuracy measure.** The three fully-annotated test files contain 1757 A and B chirps. We consider two metrics to explain performance: point-wise and event-wise accuracy. Point-wise accuracy is the proportion of all time points that are classified correctly. It is computed by comparing each labeled time point with the corresponding prediction, and is analogous to conventional measures of segmentation accuracy.

Event-wise accuracy, in contrast, is a more realistic assessment of DISCO's performance from the user's perspective—it measures the ability of the ensemble to detect individual sound events, and only considers runs of contiguous time points assigned to the same class. Specifically, an event was defined as a region of contiguous classifications. Extremely short events were removed by establishing minimum length threshold for events: 10 spectrogram time units, or 0.04 seconds (0.04s is 80% of the shortest chirp in the test set). Implementation of such a threshold is a simple matter of post-processing results from DISCO or any other tool. A predicted noise event is then considered a true positive if it lies within a labeled region of the same class and is above the minimum length. False positives are defined as predicted events that do not reside in a labeled region of the same class.

Both point and event-wise accuracy rely on ensemble predictions for each time point in the input spectrogram. Each prediction is associated with an IQR value describing uncertainty. We tested the performance of the ensemble as a function of IQR by considering the full range of possible IQR thresholds in a precision-recall plot, and converting above-threshold IQR predictions into the background class. Increasing the IQR threshold causes more suspect time points to be labeled as A or B, decreasing the precision and increasing recall.

Precision for event-wise accuracy is a measure of the reliability of a predicted label, and is defined as

$$\frac{\text{true positives}}{\text{true positives} + \text{false positives}}$$

while recall is a measure of the model's ability to identify all instances of a class, and is defined as

$$\frac{\text{true positives}}{\text{true positives} + \text{false negatives}}$$

More restrictive IQR thresholds reduce the number of chirp classifications (both correct and incorrect), resulting in the observed precision/recall curve. Current analysis tools (like SongExplorer or DeepSqueak [15, 16]) do not provide an option to select classification thresholds based on uncertainty estimates.

**4.1.2 Accuracy in beetle chirp data.** *4.1.2.1 Event-wise performance.* DISCO's event-wise performance on the test set is shown in Fig 5. DISCO was run with default parameters, including the default ensemble configuration (random init, 10 member ensemble). The plot was produced by varying IQR thresholds from 0 to 1—as the threshold is allowed to grow, an

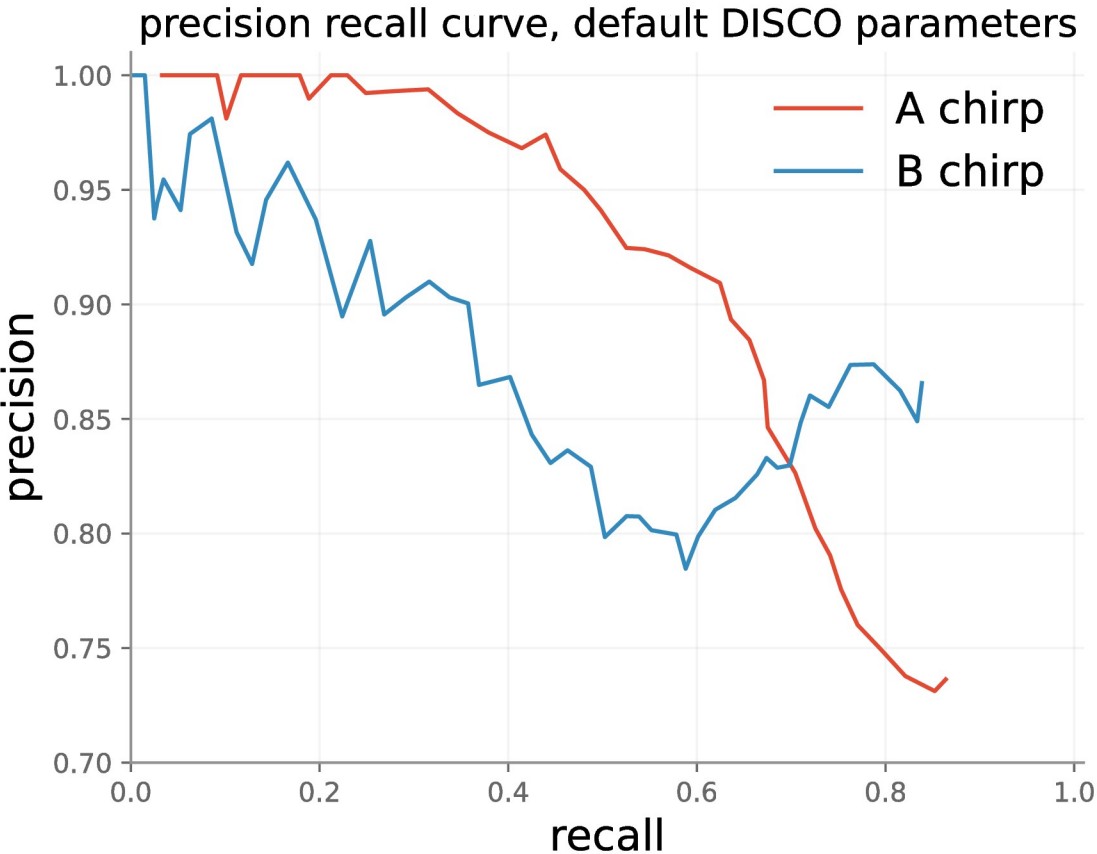

**Fig 5. DISCO's performance on the test set with default parameters.**

increasing number of sound events (true and false) are labeled. This produces a monotonic increase in recall, but a fluctuating measure of precision. Recall does not reach 100% for A or B chirps because in some cases the ensemble incorrectly classifies A or B chirps as background with high confidence.

Though these precision results at high recall in noisy data are encouraging, we suggest that they may substantially overstate concerns of false prediction. As discussed in Methods, test data were labeled based on spectrogram only, and chirps were only labeled when the annotator was confident in non-background classification. The byproduct of this conservative labeling is that many actual chirps have likely gone unlabeled in the test set because they did not visually stand out from background. To gain better insight into this possibility, we manually evaluated dozens of predicted A and B labels deemed to be incorrect by the benchmark, by both reviewing the spectrogram and listening to the corresponding sound block. In this review, we found that a majority of the "false positives" do in fact appear to be chirps. In these cases, the spectrogram typically presents a visually-ambiguous event that (with the hint of the network) appears to be a plausible sound, and the audio clip presents a sound that convincingly matches a chirp. Fig 6 presents two such examples of cases in which the signal is visually ambiguous, but (with the hint of the network) the classification of the ensemble is confirmed by listening to the recordings. The A chirps in Fig 6 either look unlike typical A chirps (subplot on the left) or are surrounded by enough background noise to be partially obscured.

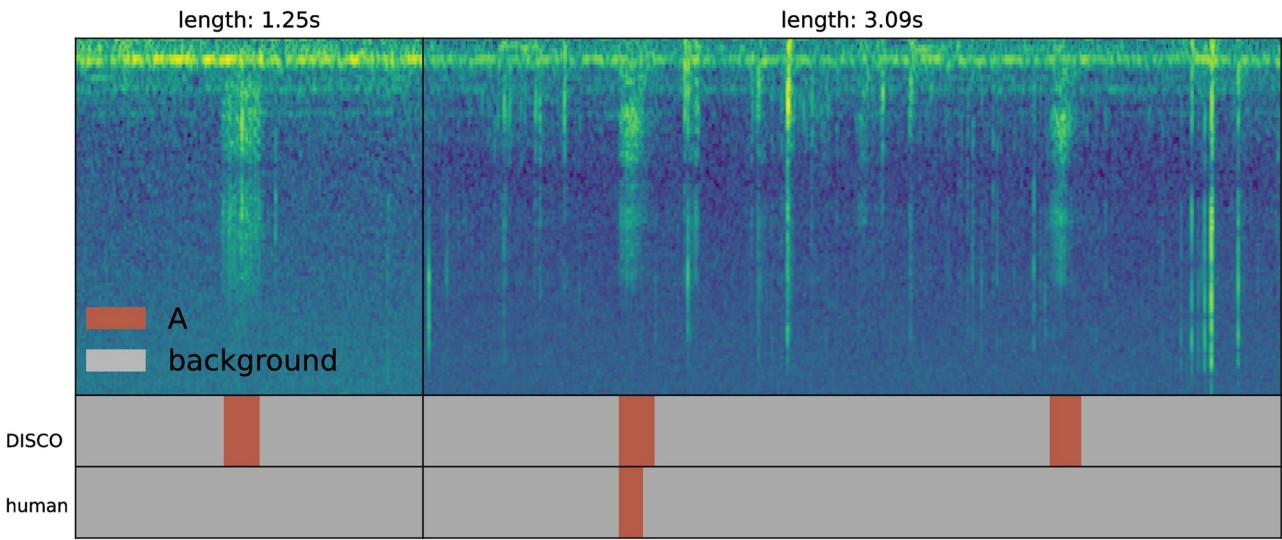

**Fig 6. Two examples of chirps identified as "false positives" in our analysis—They were called A chirps by DISCO, but labeled background by the human annotator.** The right sub-plot contains a large amount of background beetle movement noise, obscuring the desired signal. Evaluation of the source audio for these two "false positives" confirms that they do, indeed, appear to be A-type chirps.

*4.1.2.2 Ensemble technique.* We explored the influence of alternative approaches to providing ensemble members with initialization and training data, with results displayed in Fig 7. The figure presents a record of ensembles with 10 members each. The bootstrapped ensembles appear to perform better than the random init ensembles on A chirps, and the inverse for B chirps. The discrepancy in A chirp performance between ensembles is most likely explained by the label uncertainty discussed above. Randomly initialized ensembles ingested more unique A chirps, leading to more diverse representations of the A class. Bootstrapped ensembles, in contrast, could learn more precise representations because of the smaller set of A chirps available. The randomly initialized ensembles do better than bootstrapped ones on B chirps since B chirps contain less label noise in the training set.

*4.1.2.3 Point-wise performance, ensemble size.* We evaluated pointwise accuracy as a function of both ensemble type and number of ensemble members, with results presented in Fig 8. All permutations perform similarly for B chirps. On A chirps, ensembles of 10 and 30 members perform better than ensembles with 2 members. This agrees with observations in literature: more ensemble members typically yield superior performance. The differences in performance between A and B chirps are partially explained by the difficult nature of collecting ground truth labels for chirps. A chirps feature a slowly decaying tail that merges into background noise, so that determining a precise end to A chirps can be guesswork.

Event-wise metrics in Fig 7 are almost 20 points better than point-wise metrics in Fig 8, because event-wise results are not influenced by minor point-wise errors. In general, we recommend using a 10-member ensemble as it balances performance increases and compute requirements.

## 4.2 Performance with introduction of artificial noise

To demonstrate that DISCO works under a variety of sound conditions, we artificially added noise to the beetle test set and calculated performance. Gaussian noise with mean 0—often called "white noise"—was added to the pre-spectrogram waveform at different signal-to-noise

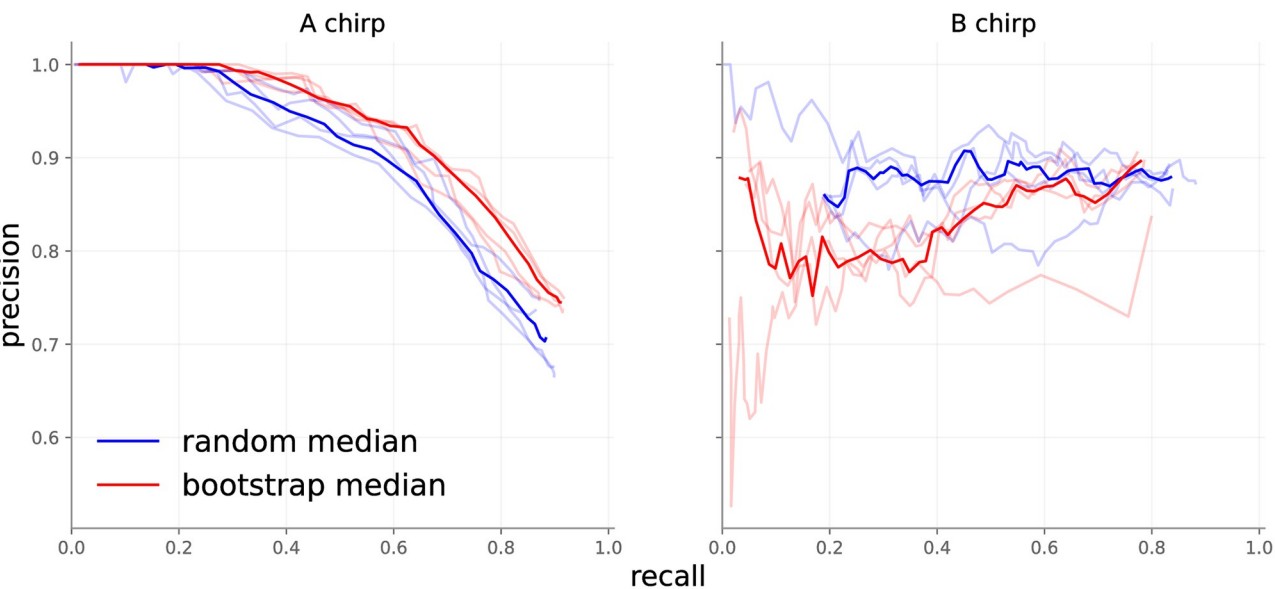

**Fig 7. 10-member ensembles with different initialization techniques.** Each experiment was replicated four times (transparent lines), and the median value of the predictions for each recall point were computed (solid lines).

ratios (SNRs). SNR is a measure that compares the strength of a signal to the strength of background noise, specifically the ratio of the power of the signal to the power of the noise. We take the mean square of an audio signal as its power in Eq 1 [31].

$$\text{SNR}_{dB} = 10\log_{10} \frac{RMS^2_{\text{signal}}}{RMS^2_{\text{noise}}} \tag{1}$$

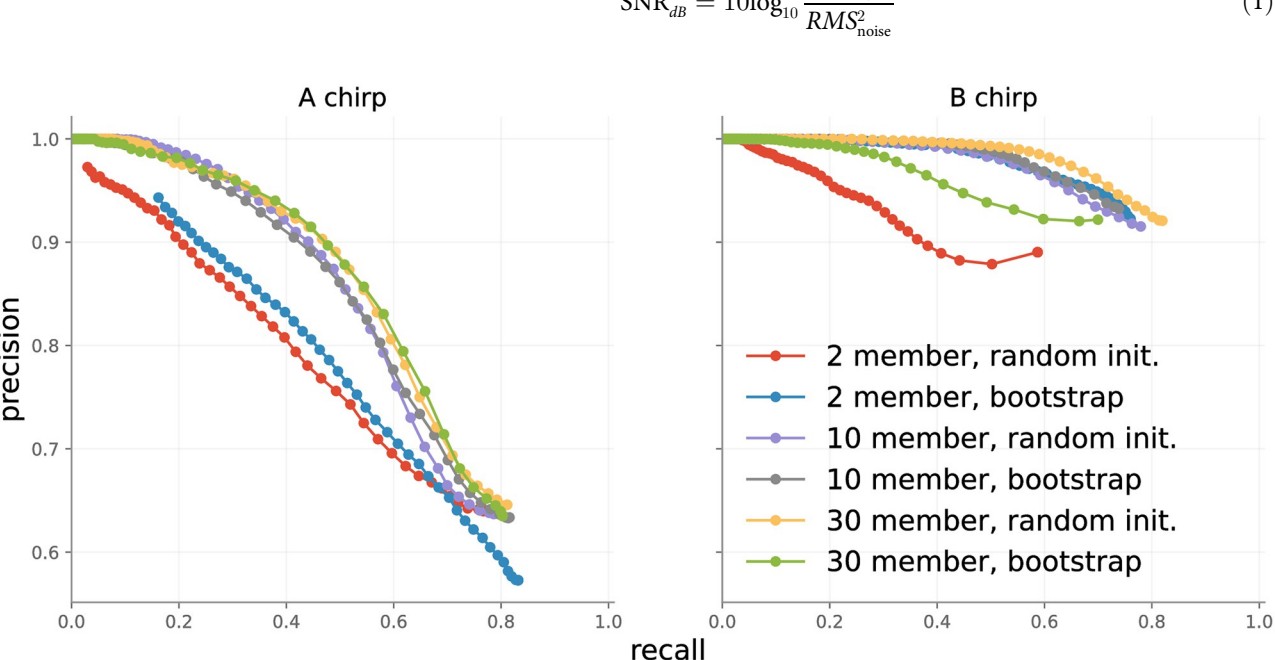

**Fig 8. Point-wise accuracy of different ensembles on the test set.**

To generate noise as specific signal-to-noise ratios (in dB), we use the following equation:

$$RMS_{\text{noise}} = \sqrt{\frac{RMS^2_{\text{signal}}}{10^{\text{SNR}_{dB}/10}}} \tag{2}$$

Since the standard deviation of Gaussian noise with mean zero is the same as the root-mean-square, we can generate Gaussian noise at specific SNRs by substituting *RMS* in Eq 2 into the formula for a Gaussian. Generated noise is then added pointwise to the input signal at different SNRs. The initial data already contains a significant amount of background noise from the sensitivity of the microphone, so that our SNR values are surely overestimated. At SNR = 50, there is little added noise distinguishable from the original recordings. When SNR = 30, chirps are barely perceptible to the human ear.

Fig 9 shows spectrograms computed from recordings with different amounts of added noise. At very low signal-to-noise ratios, the underlying signals in the uncorrupted recording are all but invisible. As SNR grows, the signals become more visible.

Fig 10 shows the performance of a randomly initialized 10-member ensemble on the same test set with different levels of noise. The flat high-recall regions in both figures correspond to cases of high SNR (low relative noise). Performance drops precipitously at about the point that human identification of sounds becomes impossible.

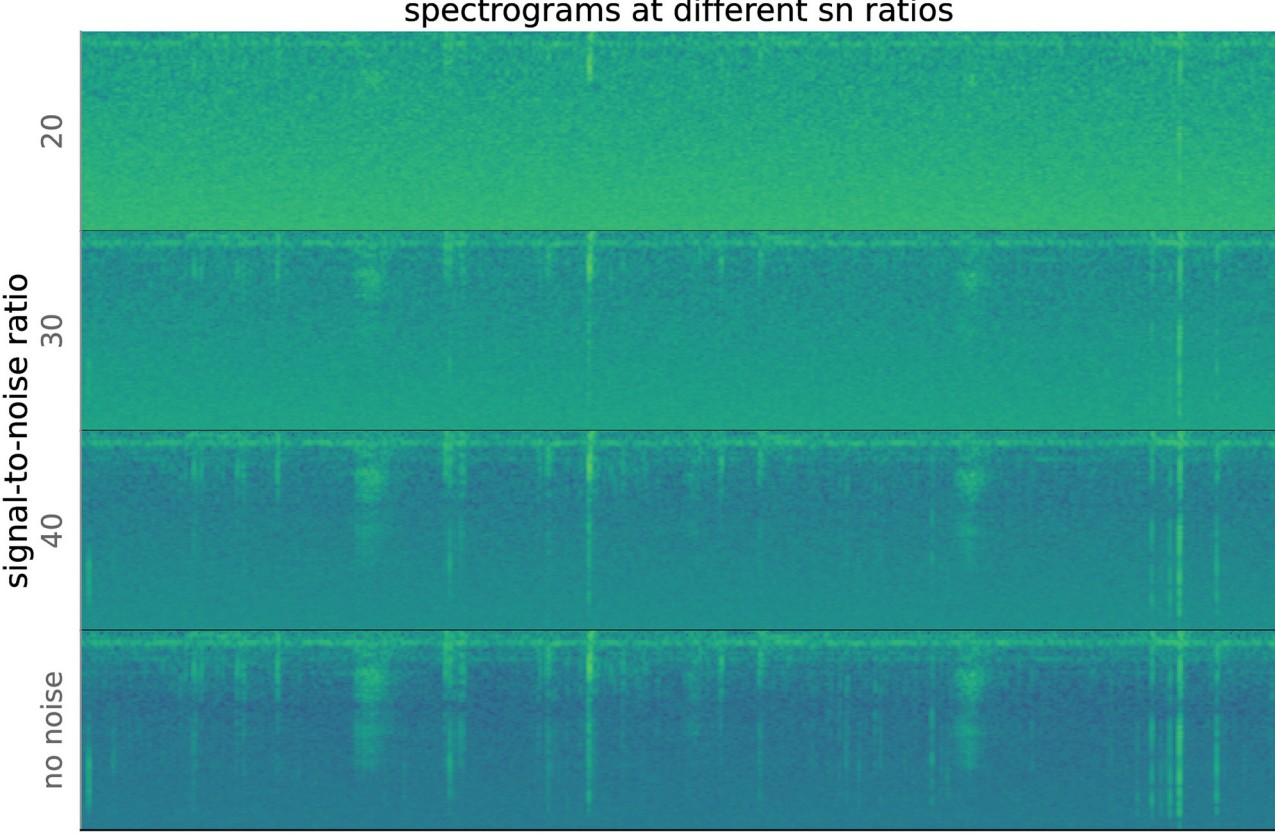

**Fig 9. 10 member random init ensemble performance with different levels of noise.** As signal-to-noise (SNR) decreases, the synthetically generated white noise increasingly interferes with recognition of sound events.

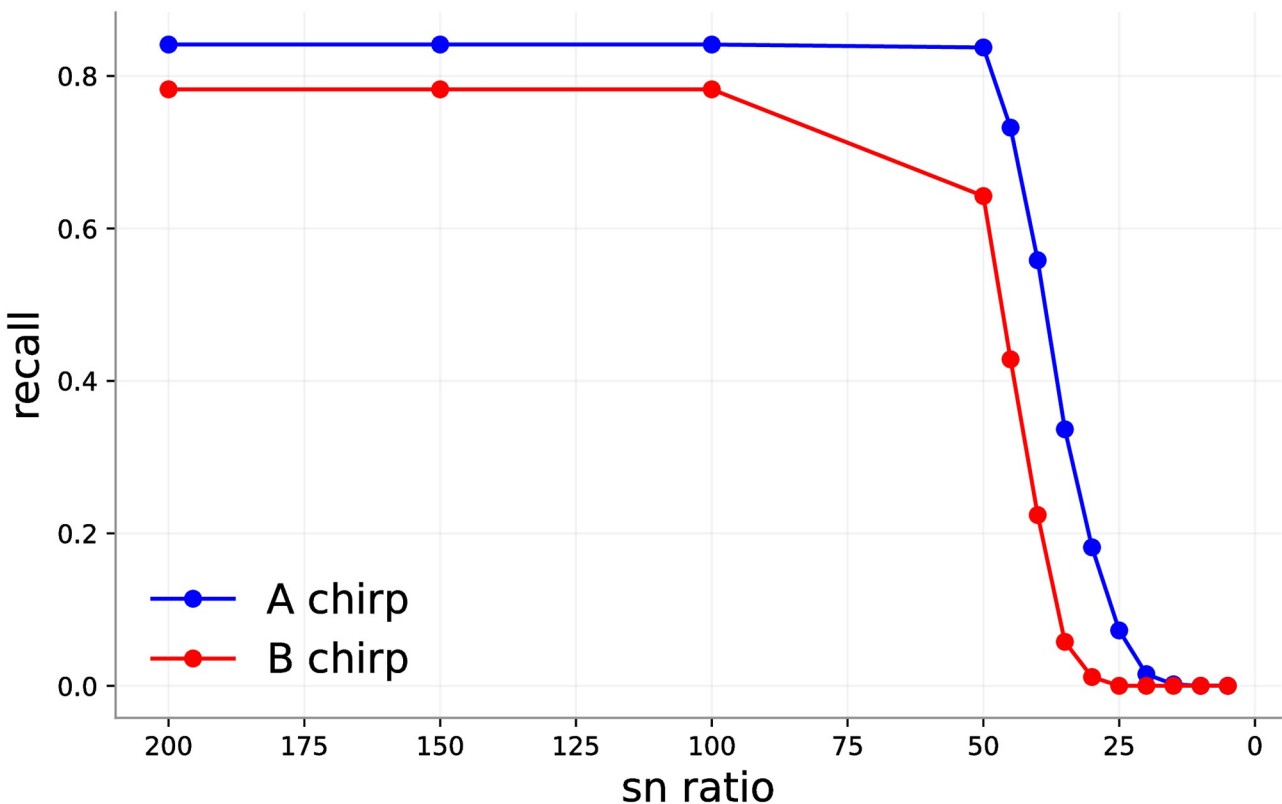

**Fig 10. Spectrograms computed from waveforms with different signal-to-noise ratios (SNRs).** As SNR decreases, the sounds in the recording become less visible.

In the noise experiment, the maximum median classification was selected, with no IQR filtering (effectively IQR threshold = 1). Not surprisingly, labeling uncertainty grows with increasing noise levels, as shown in Fig 11. The trend reverses at very high noise levels (SNR = 15 in Fig 11). IQRs closer to 0 dominate, as the ensemble is certain most points are background. At moderate levels of noise—SNR near 30, the limit of the our ability to hear the chirps—IQR distributions demonstrate longer tails, indicating higher overall uncertainty.

### 4.3 Impact of hidden Markov model smoothing

Fig 12 contains examples of the results of applying the smoothing hidden Markov model (HMM) after ensemble-based labeling. In the first subplot, a small background prediction in a long region of B predictions is converted to the correct class. In the second subplot, a unrealistic transition is correct. In the third, a low-confidence A chirp prediction is converted to background. The HMM smooths small interruptions in predictions. We observed that it occasionally smooths true positive predictions that are smaller than the minimum length threshold. This often happens with quiet or low-fidelity chirps, where only a small region is clearly visible to the ensemble. One benefit of this smoothing is that it removes very short sound elements that may pose a problem for downstream analyses. We also highlight that the precise boundaries of each event are quite difficult to establish; in these cases, each manually-

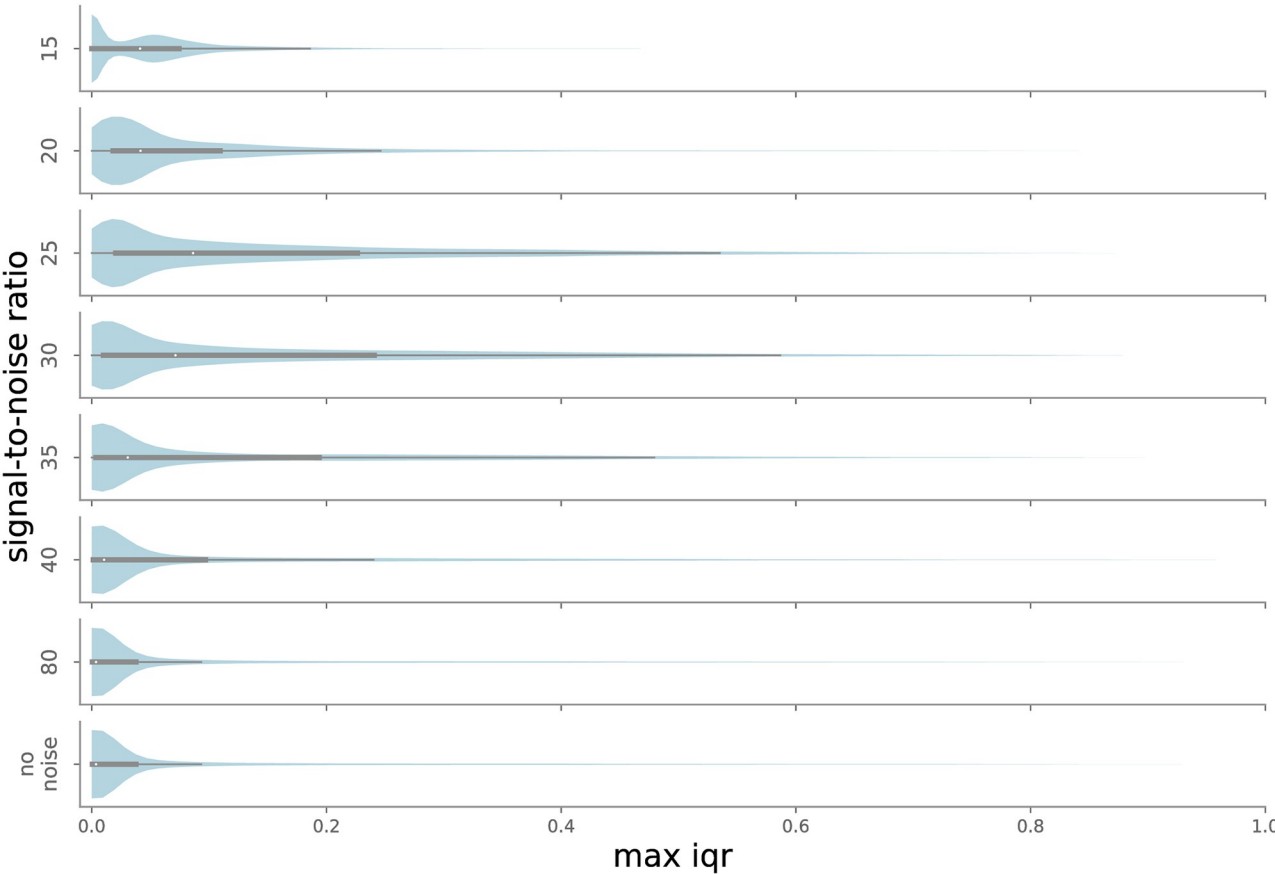

**Fig 11. Distribution of maximum IQR values for each point in the test set, as a function of added noise.** Maximum IQR values are low when all models agree on classification, i.e. when noise is low (SNR > 30, so true chirps are fairly easy to identify) or very high (SNR < 20, so noise drowns out almost all chirps). For intermediate SNR values, some ensemble members may recognize a chirp while others do not, leading to higher typical IQR values.

selected label has a range that disagrees with the post-HMM range, which itself appears to be plausibly the correct range.

### 4.4 Run time

On average, DISCO models took 7 minutes on an NVIDIA Tesla V100 GPU with 32Gb of VRAM to train for 100 epochs. On the same GPU, DISCO took 65 minutes to analyze 117Gb of recordings—280 separate files containing 179 hours of data.

### 4.5 Evaluation on alternative sound type—right whale

As DISCO was designed to be a general-purpose sound annotation tool, we analyzed another sound dataset of right whale calls [32]. This is a dataset of up-calls from right whales collected by underwater microphones on buoys. The data consists of short slices of larger recordings— each slice contains either just background noise or a right whale call surrounded by a small amount of background noise. Each slice is associated with one label even though containing multiple time points (which, in the case of whale call slices, may actually belong to different classes). Because this is a binary classification problem, DISCO defaults to a simpler final

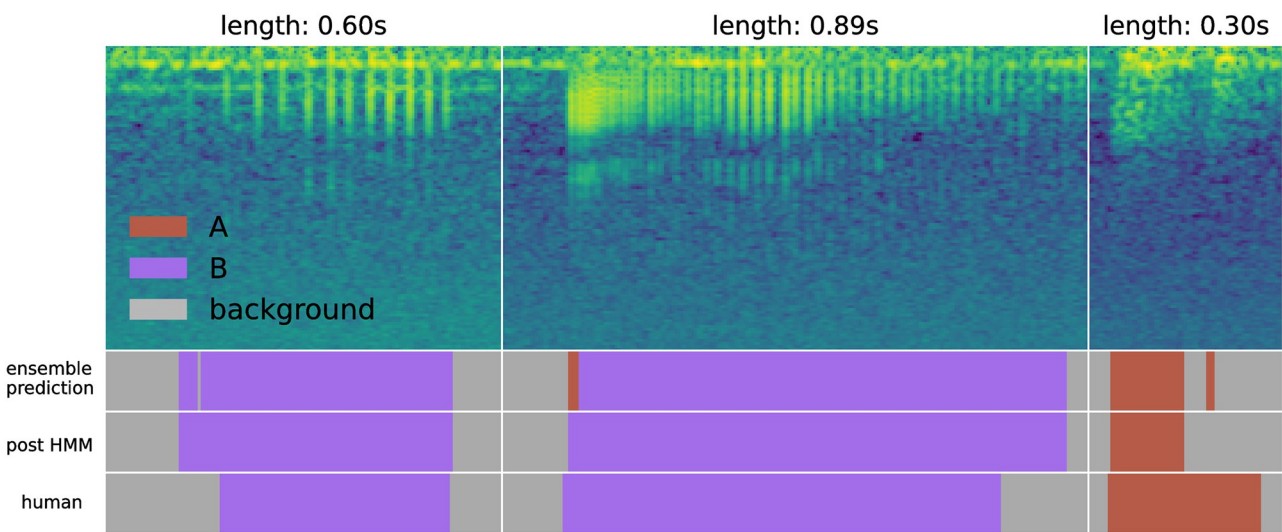

**Fig 12. Results of applying the smoothing HMM.** Small discontinuities in the predictions are converted to the correct class automatically. This figure also demonstrates the challenge of selecting correct chirp boundaries. In the two chirps on the left, the model extends the label a bit beyond the manually-selected boundary, and the choice appears to be reasonable. In the rightmost chirp, the model cuts a chirp call short; though the longer manual choice is likely the correct label, boundary ambiguity is clear.

**Table 1. Confusion matrix showing DISCO's performance on the whale dataset.**

|  |  | Predicted | |
| --- | --- | --- | --- |
|  |  | **true** | **false** |
| Actual | true | 975 | 431 |
|  | false | 188 | 4406 |

softmax layer (a single sigmoid activation for binary classification) and uses binary cross entropy as the loss function. At evaluation time, the model produces a prediction for each time-point in each short slice. We implemented a simple post-processing analysis to take the most commonly occurring prediction as the model's prediction for each sound slice. Results without hyperparameter tuning are shown in Table 1. Overall accuracy is 89%. We note that this analysis required development of 35 new lines of code to accommodate the specific format of the right whale dataset. Templates for extending DISCO in this way are available in the github repository.

Fig 13 shows a few examples of true positive, true negative, false negative, and false 430 positive classifications by DISCO on the whale data.

## 5 Conclusion

DISCO is a novel toolkit for sound classification that quantifies uncertainty and accounts for transitions between different sound classes through the application of a hidden Markov model. It includes sub-tools for labeling training data and efficiently annotating novel sound files. Using an event-based accuracy metric that closely relates to typical labeling end goals to quantify performance, we show that DISCO performs well on test datasets consisting of both beetle chirps and whale songs. DISCO also produces calibrated estimates of uncertainty via the

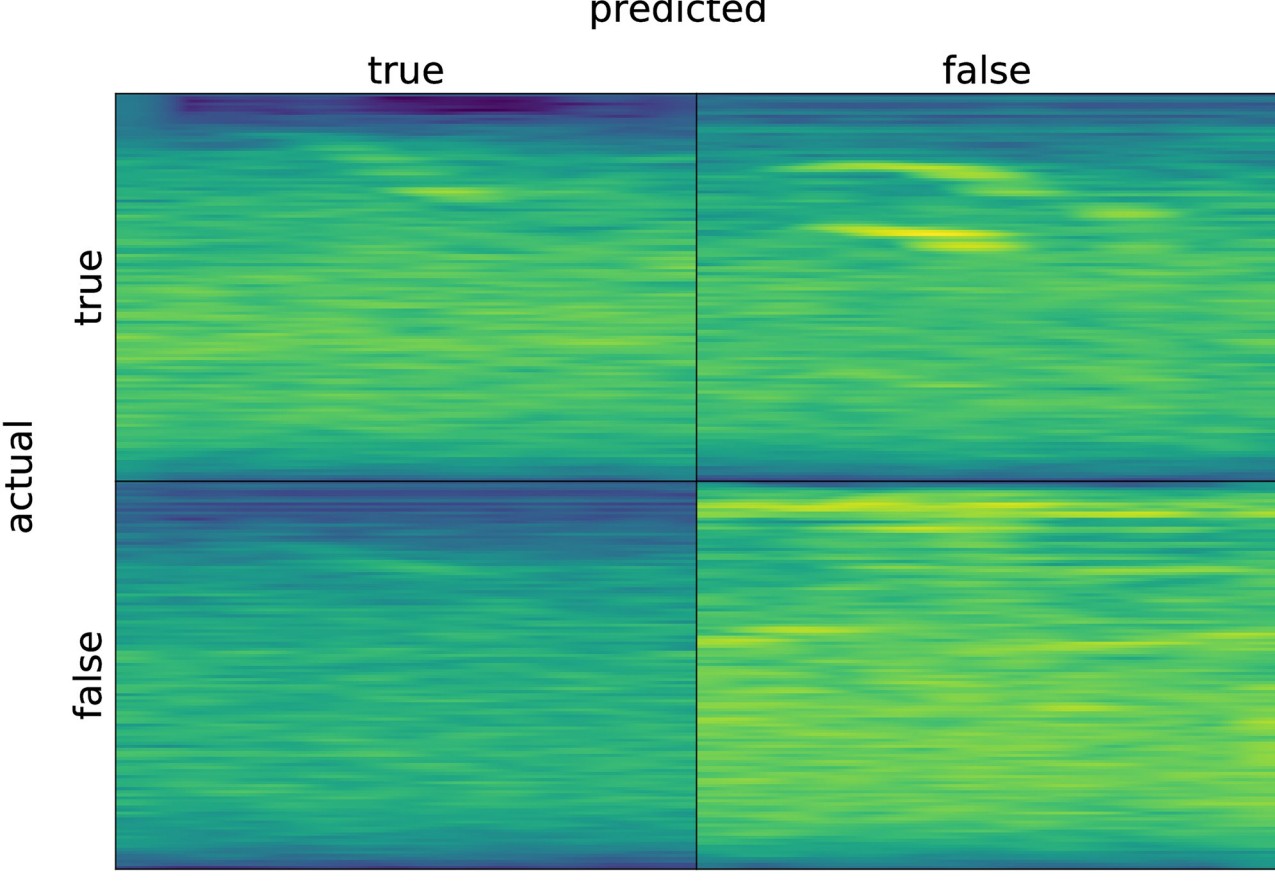

**Fig 13. Randomly-selected example spectrograms for correct and incorrect classifications by DISCO on the whale data.**

application of a model ensemble. One advantage of this feature is that it is possible to tune DISCO to output only highly-confident sound event labels. DISCO is freely available via `pip` and github, requires minimal code changes to operate on a new dataset, and is designed to be easily editable and comprehensible.

## Acknowledgments

The authors wish to thank Nathan Barton and Camille Thomas-Bulle for their guidance in acquiring, interpreting, and labeling beetle recordings. We are grateful for the use of the GSCC cluster at the University of Montana.

## Author Contributions

**Conceptualization:** Thomas Colligan, Kayla Irish, Douglas J. Emlen, Travis J. Wheeler.

**Data curation:** Kayla Irish, Douglas J. Emlen.

**Investigation:** Thomas Colligan, Travis J. Wheeler.

**Methodology:** Thomas Colligan, Kayla Irish.

**Project administration:** Travis J. Wheeler.

**Resources:** Douglas J. Emlen.

**Software:** Thomas Colligan, Kayla Irish.

**Supervision:** Travis J. Wheeler.

**Visualization:** Thomas Colligan.

**Writing – original draft:** Thomas Colligan, Travis J. Wheeler.

**Writing – review & editing:** Kayla Irish, Douglas J. Emlen, Travis J. Wheeler.

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
