## [Decision Letter · Decision Letter 0]

10 Apr 2023

PONE-D-23-03926DISCO: A deep learning ensemble for uncertainty-aware segmentation of acoustic signalsPLOS ONE

Dear Dr. Wheeler,

Thank you for submitting your manuscript to PLOS ONE. After careful consideration, we feel that it has merit but does not fully meet PLOS ONE’s publication criteria as it currently stands. Therefore, we invite you to submit a revised version of the manuscript that addresses the points raised during the review process.

We look forward to receiving your revised manuscript.

Kind regards,

Felix Albu, Ph.D.

Academic Editor

PLOS ONE

“We acknowledge funding from the National Institute of General Medical Sciences (NIGMS), National Institutes of Health (NIH) GM132600, and the Division of Integrative Organismal Systems (IOS), National Science Foundation (NSF) 2015907.”

“The authors wish to thank Nathan Barton and Camille Thomas-Bulle for their guidance

in acquiring, interpreting, and labeling beetle recordings. We are grateful for the use of

the GSCC cluster at the University of Montana. We also acknowledge funding from the

National Institute of General Medical Sciences (NIGMS), National Institutes of Health

(NIH) GM132600, and the Division of Integrative Organismal Systems (IOS), National

Science Foundation (NSF) 2015907.”

“We acknowledge funding from the National Institute of General Medical Sciences (NIGMS), National Institutes of Health (NIH) GM132600, and the Division of Integrative Organismal Systems (IOS), National Science Foundation (NSF) 2015907.”

Additional Editor Comments:

The decision is Major Revision. Please address all the comments of the reviewers in the revised paper.

Reviewers' comments:

Reviewer's Responses to Questions

**Comments to the Author**

1. Is the manuscript technically sound, and do the data support the conclusions?

Reviewer #1: Yes

Reviewer #2: Partly

2. Has the statistical analysis been performed appropriately and rigorously? 

Reviewer #1: Yes

Reviewer #2: No

3. Have the authors made all data underlying the findings in their manuscript fully available?

Reviewer #1: Yes

Reviewer #2: Yes

4. Is the manuscript presented in an intelligible fashion and written in standard English?

Reviewer #1: Yes

Reviewer #2: Yes

5. Review Comments to the Author

Reviewer #1: Introduction section must be improved

- Do not use abbreviation such as i.e. I have seen that you often use this abbreviation, so I will not repeat this advice again, it also applies to the other occurrences.

- Authors should emphasize contribution and novelty, the introduction needs to clarify the motivation, challenges, contribution, objectives, and significance/implication.

- You should introduce the problem in more detail so that the reader is immediately clear about the purpose of your study.

- Add more references to works that have already dealt with the topic (spectrogram for sound event detection), for example:” Improving smart cities safety using sound events detection based on deep neural network algorithms” and “Acoustical unmanned aerial vehicle detection in indoor scenarios using logistic regression model”

- You must properly introduce your work, specify well what were the goals you set yourself and how you approached the problem.

- At the end of the section, add an outline of the rest of the paper, in this way the reader will be introduced to the content of the following sections.

Section 1 must be improved

- Describe in detail the equipment used to make the record of beetle sound data. Extract this data from the datasheet of the instrumentation manufacturer. To make reading the specifications of the instruments more immediate, you can insert them in a table, listing the instruments used and the specific characteristics for each.

- Figure must 1 be improved: the labels of the axes are missing, as you specified the three spectrograms do not refer to the same time interval. If I had added the axes labels this would have been obvious. also the legend with the values of the color map is missing, remember to add the units of measurement

- “Beetles were recorded in 4’ x 4’ x 4’ boxes lined with anechoic studiofoam wedges to insulate the animals from ambient sounds like ceiling fans” Where are the photo of the measures? This way it would have been clearer how the experiment was set up

- 102) “Humans are better at discerning differences in lower frequencies than higher frequencies” Add references to support these statements.

- Figure must 2 be improved: the labels of the axes are missing, as you specified the three spectrograms do not refer to the same time interval. If I had added the axes labels this would have been obvious. also the legend with the values of the color map is missing, remember to add the units of measurement

Section 2 must be improved

- The section relating to the methodologies based on Machine Learning must be enriched. You must summarize the essential characteristics of the methods you have used and justify your choices. Try to summarize what are the strengths and weaknesses of the methods, in this way you can make the reader understand why you have chosen these methodologies.

- I could not find a detailed description of the evaluation metrics you have adopted. How will you measure your model's performance? This section is essential in order to demonstrate the effectiveness of your methodology. Furthermore, only by adopting adequate metrics will it be possible to compare your results with those obtained by other researchers.

Section 3 must be improved

- a description of the hardware and software used for data processing is completely missing. Describe in detail the hardware used: Extract this data from the datasheet of the hardware manufacturer. To make reading the specifications of the hardware more immediate, you can insert them in a table, listing the instruments used and the specific characteristics for each.

- Also, you should describe in detail the software platform you used.

- Finally describe the machine learning-based libraries you used.

- A detailed description of the sample size on which the model was trained is missing. How much data did you have available? How did you split them between training and testing?

- How did you set the parameters of the algorithm?

- Did you perform a hyperparameter optimization?

Section 4 must be improved

- Paragraphs are missing where the possible practical applications of the results of this study are reported. What these results can serve the people, it is necessary to insert possible uses of this study that justify their publication.

- They also lack the possible future goals of this work. Do the authors plan to continue their research on this topic?

Reviewer #2: The paper needs a lot of improvement.

Quality content needs to be added to the work.

Proper comparative analysis and research gaps for studied domain should be explained

There should be a detailed explanation of the results computed and analyzed.

Improvise the abstract and conclusion section.

Grammatical mistakes should be removed from the work.

Literature review related to work further comparative analysis needs to be added to the paper.

Add more references and convert paper in proper format.

6. PLOS authors have the option to publish the peer review history of their article (what does this mean?). If published, this will include your full peer review and any attached files.

Reviewer #1: No

Reviewer #2: No

---

## [Author Response · Author response to Decision Letter 0]

31 May 2023

To Whom It May Concern:

Following the recommendations of the editor, we have updated author information and removed funding information from the Acknowledgements section. We would like to ensure that the submitted Funding Statement now reads: 

"We also acknowledge funding from the National Institute of General Medical Sciences (NIGMS), National Institutes of Health (NIH) GM132600, and the Division of Integrative Organismal Systems (IOS), National Science Foundation (NSF) 2015907. The funders had no role in study design, data collection and analysis, decision to publish, or preparation of the manuscript."

In addition, we have addressed reviewer comments as described in the text below. In the revised manuscript, notable changes in text have been highlighted in a red font. In this response letter, reviewer comments and our responses are captured in order of appearance within the review

Reviewer #1: 

- Do not use abbreviation such as i.e. I have seen that you often use this abbreviation, so I will not repeat this advice again, it also applies to the other occurrences.

This is a matter of preference. Though there exist many websites expressing an opinion on the use of such abbreviations, there seems to be no consensus, and we have seen no guidance from any of the journals that we regularly read. We spent under one minute scanning through abstracts in Science, and quickly found “e.g.” used twice in a single abstract (https://www.science.org/doi/10.1126/scitranslmed.abq4473). During the same brief search, we did not find “for example” in any of the skimmed manuscripts. We have used the abbreviations “i.e.” and “e.g.” for decades, and never before had a reviewer comment on their use. We have also observed their frequent use by others throughout the scientific literature. If the editors of this journal instruct us to deviate from our preferred style, we will do so.

~~~~

- Authors should emphasize contribution and novelty, the introduction needs to clarify the motivation, challenges, contribution, objectives, and significance/implication.

- You should introduce the problem in more detail so that the reader is immediately clear about the purpose of your study.

We have made modifications to the Introduction section that we believe address this comment.

 - The manuscript introduces the general motivation (analysis of recordings of animal sounds) and the more specific motivation (accurate delineation and classification of sound events, improved ease of model training, novel presentation of classification uncertainty)

 - It specifies the objectives (“ease of training, automate the process of classifying sound events, provide accurate estimations of classification uncertainty”)

 - We have added a sentence indicating the anticipated significance: enabling scientists to produce improved automated sound event classification. 

~~~~

- Add more references to works that have already dealt with the topic (spectrogram for sound event detection), for example:” Improving smart cities safety using sound events detection based on deep neural network algorithms” and “Acoustical unmanned aerial vehicle detection in indoor scenarios using logistic regression model”

The manuscript already cites the appropriate work. Both SongExplorer and DeepSqueak compute spectrograms in the first aspect of their analysis, then employ alternative approaches thereafter. The works suggested by the reviewer seem to be either a strange choice of disconnected context (cities and UAVs?) or an instance of citation coercion. In either case, we do not believe that addition of these citations is warranted, given the biological focus of this paper.

- You must properly introduce your work, specify well what were the goals you set yourself and how you approached the problem.

This appears to be a repeat of the earlier comment beginning with “Authors should emphasize…”. We believe that we have done so.

~~~~

- At the end of the section, add an outline of the rest of the paper, in this way the reader will be introduced to the content of the following sections.

This is a matter of preference. We are aware that some authors like to include such text in their Introduction sections, but we find such sections to be non-informative. The omission of this sort of text is intentional.

~~~~

- Describe in detail the equipment used to make the record of beetle sound data. Extract this data from the datasheet of the instrumentation manufacturer. To make reading the specifications of the instruments more immediate, you can insert them in a table, listing the instruments used and the specific characteristics for each.

The manuscript already includes quite a bit of detail regarding recording equipment and conditions: 

“… recorded in 4’ x 4’ x 4’ boxes lined with anechoic studiofoam wedges to insulate the animals from ambient sounds like ceiling fans. They were recorded with a Sennheiser ME62 omnidirectional microphone and K6 power module. Recordings were single-channel and had a sample rate of 48Khz …”

Considering that this manuscript is introducing software for general sound event classification, and not producing any particular biological analysis of the animals being recorded, we believe that inclusion of any further details would be both unnecessary and distracting.

~~~~

- Figure must 1 be improved: the labels of the axes are missing, as you specified the three spectrograms do not refer to the same time interval. If I had added the axes labels this would have been obvious. also the legend with the values of the color map is missing, remember to add the units of measurement

We have included a duration axis label for each sound recording, and added text to the figure caption that better explains the meaning of a spectrogram.

~~~~

- “Beetles were recorded in 4’ x 4’ x 4’ boxes lined with anechoic studiofoam wedges to insulate the animals from ambient sounds like ceiling fans” Where are the photo of the measures? This way it would have been clearer how the experiment was set up

Considering that this manuscript is introducing software for general sound event classification, and not producing any particular biological analysis of the animals being recorded, we believe that addition of a photograph of the measurements would be both unnecessary and distracting.

~~~~

- “Humans are better at discerning differences in lower frequencies than higher frequencies” Add references to support these statements.

Thank you for catching this oversight. We have added references to:

 Oxenham. “How we hear: The perception and neural coding of sound”

and

 Stevens, et al. “A scale for the measurement of the psychological magnitude pitch”.

~~~~

- Figure must 2 be improved: the labels of the axes are missing, as you specified the three spectrograms do not refer to the same time interval. If I had added the axes labels this would have been obvious. also the legend with the values of the color map is missing, remember to add the units of measurement

It looks to us like the reviewer copied their earlier comment, without editing to relate to the details of this figure. For example: both images in this figure represent the same duration. We are content with the labels provided, and believe that extra labeling is superfluous.

~~~~

- The section relating to the methodologies based on Machine Learning must be enriched. You must summarize the essential characteristics of the methods you have used and justify your choices. Try to summarize what are the strengths and weaknesses of the methods, in this way you can make the reader understand why you have chosen these methodologies.

We struggle to understand what the reviewer imagines we should do differently. Our text most definitely describes our methods, motivations for using them, and the strengths and weaknesses. We have made no adjustments in responds to this comment.

- I could not find a detailed description of the evaluation metrics you have adopted. How will you measure your model's performance? This section is essential in order to demonstrate the effectiveness of your methodology. Furthermore, only by adopting adequate metrics will it be possible to compare your results with those obtained by other researchers.

We refer the reviewer to section 3.1.1 (“Data input, accuracy measure”), where they can find a description of the basic evaluation method that we have adopted. Each subsequent subsection within Results provides a description of the analysis performed and measures used for various analyses. 

~~~~

- a description of the hardware and software used for data processing is completely missing. Describe in detail the hardware used: Extract this data from the datasheet of the hardware manufacturer. To make reading the specifications of the hardware more immediate, you can insert them in a table, listing the instruments used and the specific characteristics for each.

We refer the reviewer to section 3.4, which describes the hardware used for evaluation. Since timing is the only aspect of this analysis with a dependency on specific hardware, we find this to be entirely sufficient. We have never seen a paper present a tabular representation of a computer’s hardware datasheet, and have no interest in being the first to do so.

~~~~

- Also, you should describe in detail the software platform you used.

We are not sure, but we think the reviewer is suggesting that we describe the operating system on which experiments are run. Our response is that this is irrelevant to the analysis at hand. DISCO is written in Python with universal libraries, so is agnostic to operating system. If the suggestion instead is that we list all libraries, we refer to the response to the next comment, and note DISCO is open source and available on github, meaning that users/reviewers are welcome to inspect the code for further detail.

v

- Finally describe the machine learning-based libraries you used.

We have added mention of the three primary ML library dependencies (Torchaudio, PyTorch, and pomegranate) in the Software Engineering section.

~~~~

- A detailed description of the sample size on which the model was trained is missing. How much data did you have available? How did you split them between training and testing?

We refer the reviewer to section 2.3 (“Analysis setup”), which provides this information.

~~~~

- How did you set the parameters of the algorithm?

- Did you perform a hyperparameter optimization?

We have added text to section 2.3 to address these comments.

~~~~

- Paragraphs are missing where the possible practical applications of the results of this study are reported. What these results can serve the people, it is necessary to insert possible uses of this study that justify their publication.

The manuscript presents two example applications of the software (labeling sound events in beetle songs and whale songs), introduced the software in the context of the more general problem of sound event classification, and closed with a reminder that the software can be easily generalized to other data sets. We believe that any expansion on these, or repetition of the ways that DISCO can be applied, would feel redundant and distracting.

~~~~

- They also lack the possible future goals of this work. Do the authors plan to continue their research on this topic?

This is a stylistic preference. Though we sometimes include such text in our manuscripts, we do not agree that indication of future plans is a requisite part of a manuscript.

~~~~~~~~~~~~~~~~~~~~

~~~~~~~~~~~~~~~~~~~~

Reviewer #2: 

Version 1 of the review (listed here) was not actionable; we requested clarification.

The paper needs a lot of improvement.

Quality content needs to be added to the work.

Proper comparative analysis and research gaps for studied domain should be explained

There should be a detailed explanation of the results computed and analyzed.

Improvise the abstract and conclusion section.

Grammatical mistakes should be removed from the work.

Literature review related to work further comparative analysis needs to be added to the paper.

Add more references and convert paper in proper format.

 ~~~~

Comments from version 2 of the review are listed here

- The text jumps right into discussing sound data and the spectrogram without providing much context for why this information is important or how it is used.

We have made modifications to the Introduction section that we believe address this comment.

~~~~

- Explain the purpose and significance of the U-Net architecture, as well as its limitations and benefits in the context of the study.

- Clarify the differences between the 1-D and 2-D U-Net architectures, and why the 1-D U-Net is better suited for temporal segmentation.

We have made modifications to section 2.1 to address this comment.

~~~~

- Explain the advantages of using an ensemble of classification models over a single model, and how this improves accuracy and reduces uncertainty in the results.

We have made modifications to section 2.2 to address this comment.

~~~~

- Provide more information on the dataset used for the study, including its size, sources, and other relevant details.

We refer the reviewer to section 2.3 (“Analysis setup”), which provides this information.

~~~~

- Provide more information on the evaluation metrics used to assess the performance of the model ensemble, and how they relate to the objectives of the study.

We refer the reviewer to section 3.1.1 (“Data input, accuracy measure”), where they can find a description of the basic evaluation method that we have adopted. Each subsequent subsection within Results provides a description of the analysis performed and measures used for various analyses. In each case, the manuscript provides a motivation for the analysis, presents the relevant calculations, and explains the results.

~~~~

- Explain the rationale for using different ensembling techniques (bootstrap aggregation and random initialization), and how they differ in their approach to addressing the challenges of sound labeling.

The differences between these two ensembling strategies is described in the 2nd paragraph of section 2.2 (“An ensemble of classification models”). We have added a phrase to highlight that we evaluated both in order to identify the utility of the alternative options. 

~~~~

- Clarify how uncertainty is estimated in the ensemble, and how it is used to make predictions and improve the accuracy of the results.

We refer the reviewer to section 2.2 (“An ensemble of classification models”), which provides this information.

~~~~

- Consider including more examples of chirp data and how DISCO processes it.

We did consider this, then opted not to do so, since we found that the extra examples were distracting and provided little extra insight into DISCO’s performance or behavior. The examples we have provided give a good sense of the diversity of inputs and complications. 

~~~~~~~~~~~~~~~~~~~~~~~~~~~~

~~~~~~~~~~~~~~~~~~~~~~~~~~~~

We hope that these are all welcome changes and that the improved manuscript satisfies the quality and formatting guidelines for publishing with PLOSOne.

---

## [Decision Letter · Decision Letter 1]

22 Jun 2023

DISCO: A deep learning ensemble for uncertainty-aware segmentation of acoustic signals

PONE-D-23-03926R1

Dear Dr. Wheeler,

We’re pleased to inform you that your manuscript has been judged scientifically suitable for publication and will be formally accepted for publication once it meets all outstanding technical requirements.

Kind regards,

Felix Albu, Ph.D.

Academic Editor

PLOS ONE

Additional Editor Comments (optional):

The decision is Accept.

Reviewers' comments:

Reviewer's Responses to Questions

**Comments to the Author**

1. If the authors have adequately addressed your comments raised in a previous round of review and you feel that this manuscript is now acceptable for publication, you may indicate that here to bypass the “Comments to the Author” section, enter your conflict of interest statement in the “Confidential to Editor” section, and submit your "Accept" recommendation.

Reviewer #1: All comments have been addressed

Reviewer #2: All comments have been addressed

2. Is the manuscript technically sound, and do the data support the conclusions?

Reviewer #1: Yes

Reviewer #2: Yes

3. Has the statistical analysis been performed appropriately and rigorously? 

Reviewer #1: Yes

Reviewer #2: Yes

4. Have the authors made all data underlying the findings in their manuscript fully available?

Reviewer #1: Yes

Reviewer #2: Yes

5. Is the manuscript presented in an intelligible fashion and written in standard English?

Reviewer #1: Yes

Reviewer #2: Yes

6. Review Comments to the Author

Reviewer #1: (No Response)

Reviewer #2: The paper titled "Research on Automatic Detection System of Drawing Defects Based on Machine Vision" presents a comprehensive study on developing an automatic detection system for packaging paper defects. All concerns addressed by the author in detail. The work is apt and can be considered for publication. The work can be further extended in various domains.

7. PLOS authors have the option to publish the peer review history of their article (what does this mean?). If published, this will include your full peer review and any attached files.

Reviewer #1: No

Reviewer #2: No

---

## [Editor Report · Acceptance letter]

18 Jul 2023

PONE-D-23-03926R1 

DISCO: A deep learning ensemble for uncertainty-aware segmentation of acoustic signals 

Dear Dr. Wheeler:

I'm pleased to inform you that your manuscript has been deemed suitable for publication in PLOS ONE. Congratulations! Your manuscript is now with our production department. 

Kind regards, 

on behalf of

Dr. Felix Albu 

Academic Editor

PLOS ONE